# The REE-Zr-U-Th Minerals of the Maronia Monzodiorite, N. Greece: Implications on the Saturation and Segregation Mechanisms of Critical Metals in Intermediate–Mafic Compositions

Charalampos Vasilatos * and Angeliki Papoutsa *

Department of Geology and Geoenvironment, School of Science, National and Kapodistrian University of Athens, Panepistimiopolis, Zografos, 15784 Athens, Greece
* Correspondence: vasilatos@geol.uoa.gr (C.V.); angpapou@geol.uoa.gr (A.P.)

**Abstract:** This work delves into the presence of REE-Ti-Zr-U-Th minerals, in the mafic–intermediate rocks of the Maronia pluton, Greece, an Oligocene intrusion formed through arc-magmatism during subduction. In Maronia monzodiorite, critical metals are contained in three principal mineral groups, namely, the REE-Ti-Zr, REE-Ca-P, and U-Th assemblages. The REE-Ti-Zr group includes REE-ilmenite, chevkinite-like phases, zirconolite, and baddeleyite. The REE-Ca-P assemblage is represented by allanite-(Ce), monazite-(Ce), and huttonitic monazite-(Ce). The U-Th assemblage comprises thorite–coffinite and uraninite–thorianite solid solutions. The paragenetic sequencing of these minerals offers insights into their formation conditions and correlation with the pluton's magmatic evolution. In the REE-Ti-Zr group, mineral formation progresses from REE-ilmenite to baddeleyite through chevkinite-like phases and zirconolite under oxidizing conditions. The REE-Ca-P sequence involves allanite-(Ce), followed by monazite-(Ce), late allanite-(Ce), and huttonitic monazite-(Ce). In the U-Th group, earlier thorite–coffinite phases are succeeded by uraninite–thorianite solid solutions, indicating Si-undersaturation at late magmatic stages. Fluctuations in Ca-activity induce alternating formations of allanite-(Ce) and monazite-(Ce). These mineral variations are attributed to early-stage interactions between high-K calc-alkaline and shoshonitic gabbroic melts, influencing critical metal enrichment and mineral speciation. The study's insights into paragenesis and geological processes offer implications for mineral exploration in analogous geological settings.

**Keywords:** critical metals; Rare Earth Elements; monzodiorite; Maronia pluton; economic geology

## 1. Introduction

Rare Earth Elements (REE) are critical components in various industries, including green energy technologies, electronics, and defense applications [1,2]. The high demand for specific scarce elements within the REE group, such as neodymium, dysprosium, and europium, which are critical for the production of magnets, lasers, and phosphors, increases their criticality [3,4].

The criticality of REE in Europe cannot be overstated. Europe is heavily reliant on imports to meet its growing demand for REE, making it vulnerable to supply disruptions and geopolitical uncertainties [5]. The lack of sufficient domestic mineral resources puts Europe at a significant disadvantage in terms of ensuring a stable and sustainable supply of REE. Traditional REE producers, such as China, currently dominate the global market, leaving Europe in a position of dependence. The exploration and development of REE deposits in Europe are stalled by various factors, including geological complexity, regulatory hurdles, and environmental concerns.

Recognizing the strategic importance of REE, European countries and the European Union have been taking steps to address this criticality, leading to the most recent European Critical Raw Materials Act in 2023. Efforts are underway to promote REE exploration, enhance recycling technologies, and foster innovation in the development of alternative

materials and technologies that reduce reliance on REE. Additionally, initiatives are being implemented to promote international collaboration, diversify supply chains, and secure long-term access to critical raw materials, including REE. The criticality of REE in Europe highlights the urgent need for domestic exploration and exploitation of REE deposits. The limited occurrence of REE mineral resources within Europe is mainly restricted in the Nordic countries [6], which poses challenges for ensuring a stable and sustainable supply of these essential elements. To overcome this challenge, it is crucial to promote research in targeting REE-enriched formations.

Understanding the geological processes that contribute to the magmatic and hydrothermal enrichment of REE is essential for the effective exploration of these metals. Known deposits, globally, include REE hosted mainly in alkaline felsic rocks, and carbonatites, pegmatites, and hydrothermal deposits associated with Iron Oxide–Copper–Gold (IOCG) mineralization such as in the Gardar Province (Southern Greenland), Bayan Obo (in the west of Inner Mongolia, China), Strange Lake (Québec Canada), and Olympic Dam (Olympic Dam) prospects [7–11]. Alkaline granites can contain elevated concentrations of REE, particularly in the late-stage differentiates. In carbonatites and alkaline granites, REE can be enriched through magmatic processes, such as complex crystal fractionation and fluid–melt interactions [12–15]. These processes involve the separation and accumulation of specific minerals that have a higher affinity for REE, leading to localized enrichment. Hydrothermal fluids also play a significant role in the redistribution and concentration of REE. The circulation of fluids with ligands such as halogens and $CO_2$ may facilitate the leaching and mobilizing of REE from a primary source [16,17]. These fluids can then deposit REE in specific zones, resulting in hydrothermal veins or replacement deposits.

While alkaline felsic igneous rocks provide significant hosts for REE, their mafic–intermediate counterparts are generally not considered in the exploration of REE. REE deposits in gabbros and diorites are relatively uncommon compared to other rock types, and in such occurrences, the enrichment of REE is often related to late-stage hydrothermal processes [18,19]. Nevertheless, ongoing research and exploration efforts continue to enhance our understanding of REE occurrences in diverse geological settings, including gabbros and diorites.

The purpose of this work is to present new data on an REE-Ti-Zr-U-Th mineral assemblage, of which is recorded in the monzogabbro/diorite of the Oligocene Maronia pluton in northern Greece. Investigations on the processes and conditions that favor the enrichment in these elements and mineral formation contribute to a comprehensive understanding of REE mineralization processes and aid in the identification and evaluation of potential REE resources in analogous rock types.

## 2. Geological Setting

### 2.1. Regional Geology

The Hellenides comprise an accretionary orogen constructed during the Alpine-Himalayan Orogen, during which the continental fragments of Apulia and Pelagonia were separated from the African plate in Early Mesozoic and accreted onto Europe as microcontinents, creating the oceanic domains of Neotethys [20]. Distinct nappes represent these accreted continental parts, such as the Rhodope and Serbomacedonian massifs, and Pelagonian and Ionian zones, alternating with the relics of oceanic basins such as in the Circum Rhodope Belt, Vardar and Pindos Zones [20–23]. Cenozoic extensional tectonics from Eocene to Miocene resulted in the exhumation of the lower crustal rocks in Rhodope Massif and the Aegean, and the formation of metamorphic core complexes, presumably associated with a subducted slab retreat in the Aegean region [20,24–29].

The basement of the Rhodope Massif contains a sequence of amphibolite–eclogite facies metamorphic rocks that represent continental and oceanic crusts, and asthenospheric mantle, represented by amphibolites and meta-peridotites (garnet-spinel pyroxenites). These are exposed in the structurally uppermost Kimi Complex, in the underlying Sidironeron and Kerdyllion Complexes, and in the basal rocks of the Central Rhodopian

Gneiss and the Pangeon, Kardamos and Kechros complexes [30–33]. The protoliths of the basement rocks have been argued to be peraluminous arc-related, S-type granites, Permian gabbroic rocks, and oceanic carbonate platform sediments possibly related to Jurassic subduction [34–37].

The Rhodope Metamorphic Core Complex (RCC) [38–41] expands from Southern Bulgaria to Northern Greece and has been divided into the Northern Rhodope Core Complex (NRCC), exposed mainly in Bulgarian territories, and the Southern Rhodope Core Complex (SRCC) within Greece, separated by the Nestos Thrust [38,39]. The complex is bound to the north by the Maritsa Fault, to the west by the Vertiskos Thrust which marks the upper part of the Serbomacedonian Massif, and to the southwest by the Kerdylion Detachment, the latter being responsible for the exhumation of the core complex in ~40 Ma [25,31,41]. The NRCC includes four smaller domes, namely, the Chepinska, Arda, Kesebir-Kardamos and Kechros domes, whereas the SRCC is a larger triangular structure that has been segmented by a rifting event in Mid-Miocene [31,39].

The Circum Rhodope Belt (CRB) comprises low-grade, weakly metamorphosed sequences overlying the high-grade metamorphic rocks of the Rhodope Massif [42–45]. It is divided into the basal Makri and the overlying Drymos-Melia Units. The Makri Unit consists of a basal metasedimentary sequence of meta-conglomerates, meta-greywackes and quartzites, meta-carbonates, phyllites and schists, and an upper meta-volcaniclastic sequence of chlorite, talc mica schists and quartzites [43–45]. The Drymos-Melia Unit consists of greywackes, psammites, and schists. It lies tectonically over the Makri Unit and its basal part is an intermediate-to-mafic volcanic sequence, whereas its upper part consists of a flysch-resembling formation [46].

Tertiary syn-extensional calc-alkaline to shoshonitic magmatism within the RCC and CRB has been associated with the evolution of the core complex, and is manifested by several syn-kinematic intrusions such as Vrondou, Symvolon and Pangeon plutons [47–50]. The Kechros dome of the NRCC has been intruded by a series of Tertiary plutonic bodies that include the shoshonitic Maronia, Leptokaria, Kirki, Kassiteres, Halasmata and Tris Vryses plutons [51,52]. These intrusions have been collectively termed as the Maronia Magmatic Corridor and are considered to represent the final event of syn-extensional magmatism before its migration to the Central Aegean [53].

*2.2. Geology of Maronia Pluton*

The Maronia Pluton is located within the CRB and intrudes the basement rocks of the Makri Unit [53–55]. The pluton (Figure 1) consists of: (a) a basic group of gabbros, (b) an intermediate group of monzogabbro, quartz monzogabbro–monzonite with additional quartz monzonite, and (c) an acid group of granite and aplitic dykes and late microgranite [51,54]. From the limited exposure of the plutonic rocks in the Maronia pluton, it is inferred that the basic group occurs in the central part of the pluton, with no evident contacts within the intermediate group, whereas the granites appear as younger dykes intruding the previous lithologies. A narrow zone of contact metamorphism surrounds the pluton (Figure 1).

The almost overlapping emplacement ages from zircon at 29.7 Ma and the fission-track ages from apatite at 29.3 Ma suggest a rather rapid uplift and unroofing of the plutons [53,56]. During this period, the active Maronia Detachment Fault accommodated the exhumation of the Kechros dome and was spatially associated with the younger Maronia microgranite, emplaced at the footwall of the detachment [57]. The foliated microgranite emplacement was synchronous to shearing along a ductile shear zone [57]. Porphyry Cu-Mo-Re-Au mineralization is related to the microgranite, with molybdenite containing extremely high Re contents [55,57].

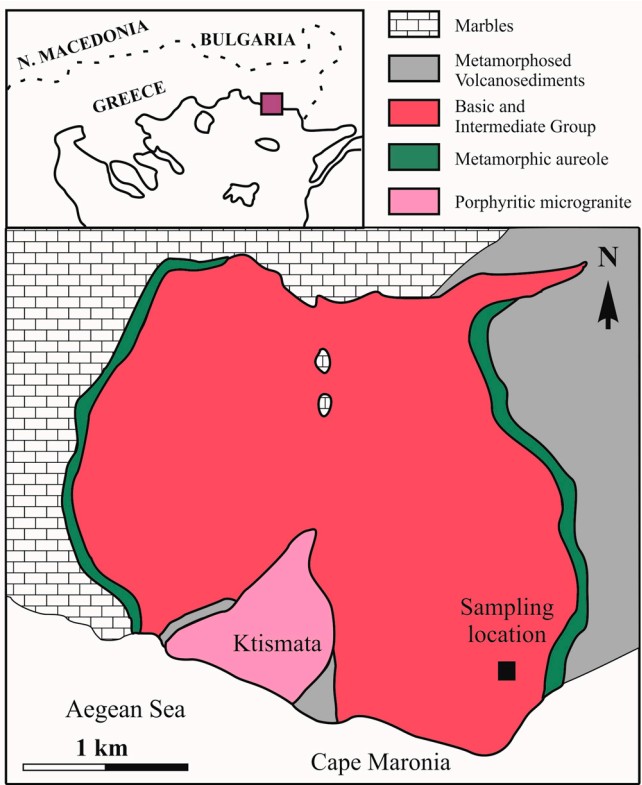

**Figure 1.** Geological map of the Maronia pluton modified from [51]. The purple box in the inset map indicates the location of Maronia pluton in northern Greece.

Although the geochemistry and petrography of major rock-forming phases is well-described [51,53,54,58], limited attention has been paid to the accessory phases hosted in these rocks [59]. Therefore, the aim of this work is to provide a thorough geochemical and mineralogical characterization of the critical metal-bearing accessory phases, correlate them to the geological evolution of the Maronia Pluton, and discuss the geological processes responsible for a primary enrichment and segregation, with implications to the mineral exploration of critical metals.

## 3. Materials and Methods

The studied samples were collected from the southeastern part of the main body of the Maronia pluton, which comprises the basic and intermediate lithological groups [51] (Figure 1). Polished thin sections of all the samples were prepared at the Institute of Geology and Mineral Exploration (IGME) in Athens. A petrographic examination of the samples was conducted at the Department of Geology and Geoenvironment, the National and Kapodistrian University of Athens.

The mineral chemistry of the accessory phases was determined through Energy Dispersive Spectroscopy (EDS) analyses at the Department of Geology and Geoenvironment, using a JEOL JSM 5600 SEM. An Oxford X ISIS 300 detector was employed, and the analytical conditions included a 20 kV accelerating beam voltage, a beam diameter of less than 2 μm, and a count time of 50 s. Calibration prior to analysis was performed using the same procedure and standards as described in [60]. For the measurements of P, La, Ce, Th, U and Zr, the standards used for calibration were GaP, $LaB_6$, $CeO_2$, $ThO_2$, U (metal) and Zr (metal), respectively. The lines used were $PK\alpha$, $LaK\alpha$, $CeL\alpha$, $ThM\alpha$, $UM\alpha$ and $ZrL\alpha$. Additional EDS analyses were performed at the Wiener Laboratory of the American School of Classical Studies in Athens. For these analyses, a JEOL JSM-IT300 InTouchScope SEM was used, equipped with an Oxford X-act Silicon Drift Detector (SDD). The same analytical conditions were applied as in the analyses conducted at the Department of Geology and Geoenvironment.

Whole-rock geochemical analyses of major and trace elements were performed at the Bureau Veritas Commodities Canada Ltd., Vancouver, Canada. The preparation of rock powders took place at the Department of Geology and Geoenvironment, where samples were crushed and subsequently pulverized using a tungsten carbide mortar. Prior to analysis, the samples were fused using lithium meta/tetraborate.

For the determination of major element concentrations, X-ray Fluorescence (XRF) was employed, with a detection limit lower than 0.01 wt.% for most elements, except for $SiO_2$ (0.1 wt.%). Inductively Coupled Plasma-Mass Spectrometry (ICP-MS) was utilized for the analysis of most trace elements, including the REE. Additionally, specific elements (Mo, Ni, Cu, Pb, Zn, As, Au, and Tl) were determined using Coupled Plasma-Optical Emission Spectrometry (ICP-OES). The whole-rock geochemical data were processed using the GCDkit software for Windows [61].

## 4. Results

### 4.1. Whole-Rock Geochemistry

A comprehensive geochemical description of the various lithologies present in the Maronia Pluton is provided in [54]. The whole-rock geochemical data obtained from the studied Maronia samples reveal a narrow range of $SiO_2$ concentrations between 52.5 and 53.2 wt.% (Table 1). Based on their whole-rock geochemistry, the samples are classified as monzodiorite (Figure 2a) and exhibit shoshonitic affinity towards high-K calc alkaline affinities (Figure 2b).

**Table 1.** Whole-rock geochemistry of the studied Maronia samples.

| Sample | MAR-1 | MAR-2 | MAR-3 | MAR-4 | MAR-5 | MAR-6 |
|---|---|---|---|---|---|---|
| Major oxides (wt.%) | | | | | | |
| $SiO_2$ | 52.9 | 52.6 | 52.7 | 52.6 | 53 | 53.2 |
| $TiO_2$ | 0.89 | 0.91 | 0.94 | 0.96 | 0.81 | 0.95 |
| $Al_2O_3$ | 16.62 | 16.5 | 16.43 | 16.49 | 16.9 | 16.33 |
| MnO | 0.16 | 0.16 | 0.15 | 0.16 | 0.16 | 0.16 |
| MgO | 4.96 | 4.99 | 4.94 | 4.97 | 4.95 | 4.96 |
| $Fe_2O_3$ | 9.37 | 9.23 | 9.26 | 9.22 | 9.12 | 9.28 |
| CaO | 8.49 | 8.28 | 8.17 | 8.25 | 8.45 | 8.16 |
| $Na_2O$ | 2.84 | 2.82 | 2.83 | 2.84 | 2.91 | 2.8 |
| $K_2O$ | 2.77 | 2.73 | 2.91 | 2.85 | 2.69 | 2.96 |
| $P_2O_5$ | 0.48 | 0.47 | 0.46 | 0.47 | 0.48 | 0.47 |
| LOI | 0.81 | 0.92 | 0.95 | 0.83 | 0.75 | 0.72 |
| Total | 100.30 | 99.62 | 99.75 | 99.65 | 100.23 | 99.99 |
| Trace elements (ppm) | | | | | | |
| Ba | 1072 | 1086 | 1045 | 855 | 1106 | 891 |
| Co | 37.5 | 38.9 | 39.8 | 42.4 | 37.5 | 40.3 |
| Ga | 17.4 | 17.3 | 17.6 | 18 | 17.6 | 18.3 |
| Hf | 4.6 | 4.5 | 5.3 | 5.2 | 5.3 | 6 |
| Nb | 10.4 | 10.9 | 10 | 11.1 | 8.8 | 11.6 |
| Rb | 122 | 125 | 130 | 135 | 122 | 139 |
| Sr | 725 | 756 | 724 | 710 | 778 | 710 |
| Ta | 0.9 | 1 | 1.3 | 1.1 | 1.1 | 1.1 |
| Th | 23.1 | 26.6 | 25.2 | 21.7 | 23 | 23.8 |
| U | 6.4 | 7.8 | 6.5 | 5.9 | 6.1 | 6.3 |
| V | 258 | 271 | 260 | 266 | 254 | 257 |
| Zr | 147 | 138 | 163 | 180 | 179 | 213 |
| Y | 22.5 | 25 | 23.2 | 24.2 | 24.3 | 24.7 |
| Mo | 1 | 1.3 | 1.2 | 1.2 | 0.8 | 0.9 |
| Cu | 96.3 | 89.8 | 87.5 | 84.2 | 71.5 | 83.5 |
| Pb | 14.2 | 9.6 | 12.7 | 10.8 | 7.6 | 9.8 |
| Zn | 41 | 40 | 43 | 42 | 41 | 39 |

**Table 1.** *Cont.*

| Sample | MAR-1 | MAR-2 | MAR-3 | MAR-4 | MAR-5 | MAR-6 |
|---|---|---|---|---|---|---|
| Ni | 19.4 | 19.5 | 19.8 | 18.6 | 17 | 17.4 |
| As | 1.4 | 1.6 | 1.7 | 1.5 | 1.6 | 1.4 |
| Cr | 37.6 | 30.8 | 34.2 | 20.5 | 34.2 | 23.9 |
| La | 33.6 | 36.3 | 36.4 | 37.5 | 37.1 | 37.5 |
| Ce | 70.2 | 74.8 | 73.9 | 78 | 77.3 | 77.4 |
| Pr | 8.91 | 9.49 | 9.26 | 9.86 | 9.65 | 10 |
| Nd | 35.6 | 39 | 37.8 | 39.2 | 38.7 | 41.3 |
| Sm | 7.18 | 7.19 | 7.51 | 7.51 | 7.44 | 7.69 |
| Eu | 1.71 | 1.85 | 1.73 | 1.71 | 1.76 | 1.72 |
| Gd | 6.07 | 6.5 | 6.33 | 6.53 | 6.57 | 6.53 |
| Tb | 0.78 | 0.77 | 0.78 | 0.8 | 0.79 | 0.82 |
| Dy | 4.73 | 4.88 | 4.75 | 4.73 | 4.38 | 4.99 |
| Ho | 0.72 | 0.78 | 0.86 | 0.86 | 0.77 | 0.9 |
| Er | 2.33 | 2.28 | 2.28 | 2.42 | 2.46 | 2.52 |
| Tm | 0.34 | 0.37 | 0.35 | 0.39 | 0.36 | 0.38 |
| Yb | 2.02 | 2.38 | 2.18 | 2.43 | 2.25 | 2.39 |
| Lu | 0.31 | 0.36 | 0.34 | 0.36 | 0.35 | 0.36 |
| ΣREE | 174 | 187 | 184 | 192 | 190 | 194 |

Notes: LOI: Loss on Ignition

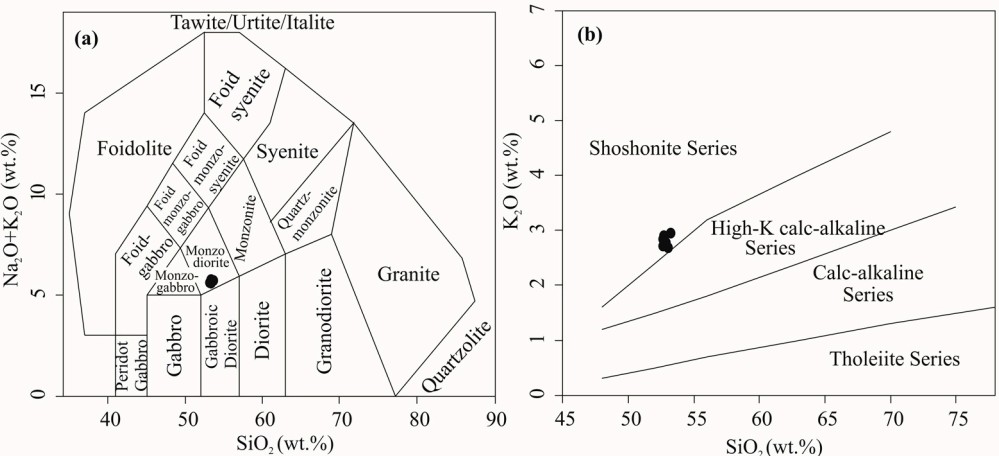

**Figure 2.** Whole-rock geochemical projection (black circles) and classification of the Maronia samples (**a**) as monzodiorite in a Total Alkalies and Silica (TAS) diagram [62], and (**b**) as rocks of shoshonitic affinity using the discrimination diagram of [63].

The concentrations of total REE in the samples range from 165 to 186 ppm (Table 1), falling within the compositional range of monzonites as reported in [53], while the concentrations of Zr range from 139 to 219 ppm (Table 1). In Maronia samples, the distributions of critical metals do not exhibit any obvious correlation with Mg-number (Figure 3a,b). On the other hand, a positive trend of REE, Y, and Zr with $TiO_2$ (Figure 3a) may be observed (r = 0.87), whereas for U and Th is weak (r = 0.76) and negative (Figure 3c,d). In all cases, any observed correlation with $TiO_2$ occurs mainly in concentrations larger than 0.9 wt.% $TiO_2$, whereas in smaller concentrations, the distribution of critical metals appears to be random.

### 4.2. Main Mineralogy of Maronia Monzodiorite

The Maronia monzodiorite samples exhibit a granitic texture that is medium- to fine-grained, occasionally displaying porphyritic features. The degree of deformations and alterations in these samples is minor. The rock-forming minerals consist primarily of homogeneous, subhedral plagioclase (60%), subhedral K-feldspar (20%), and anhedral quartz (<10%), with minor amounts of amphibole and euhedral biotite. Relics of clinopyroxene, characterized by an augite composition, can be observed. These relics exhibit disequilib-

rium textures and are replaced by amphibole (hornblende and secondary actinolite) and chlorite. Common accessory phases include epidote, apatite, titanite, and opaque minerals such as magnetite, ilmenite, chalcopyrite and pyrite.

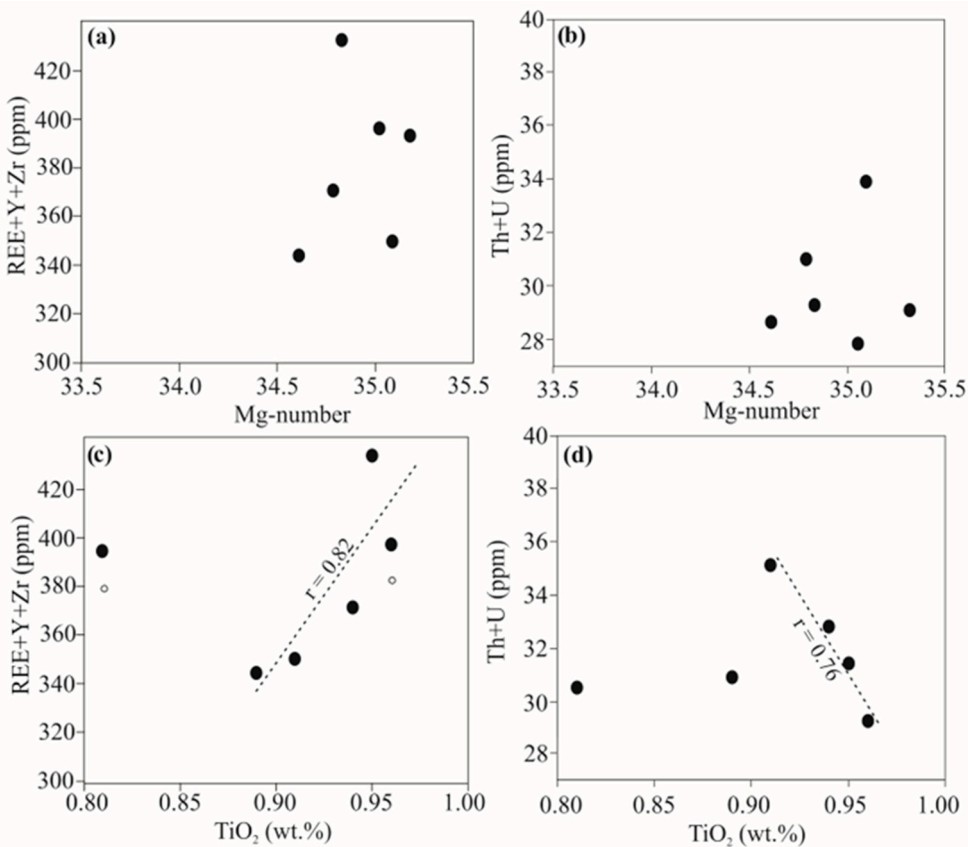

**Figure 3.** Whole-rock binary diagrams o of Maronia monzodiorite demonstrating the relationships between the concentrations of critical metals and actinides with Mg-number (**a,b**) and TiO$_2$ (**c,d**). Interpreted trend lines and correlation coefficients are presented where applicable, and exclude outliers to better depict the general trend.

The chemical composition of plagioclase in the studied samples mainly ranges from andesine to labradorite, with fewer amounts of oligoclase (Figure 4), suggesting crystallization from intermediate to mafic melt compositions. For simplicity, these studied samples will be referred to as monzodiorite, taking into consideration their whole-rock geochemical affinities as well.

### 4.3. The REE-Ti-Zr-U-Th Minerals of Maronia Monzodiorite

#### 4.3.1. Chevkinite-(Ce)-Like Phase

Chevkinite is the dominant member of the chevkinite subgroup of minerals, characterized by a general formula of A$_4$BC$_2$D$_2$(Si$_2$O$_7$)$_2$O$_8$. In this formula, the A site contains elements such as REE, Ca, Y, U, and Th; B includes elements like Fe; C consists of Al, Ti, Fe, Mn, and Zr; and the D site contains mainly Ti. Chevkinite has been reported in a wide range of igneous and metamorphic rocks, spanning from mafic to intermediate compositions, as well as peraluminous and peralkaline felsic rocks. This occurs under various pressure and temperature conditions [63,64].

Chevkinite can be distinguished chemically from its dimorph, perrierite, primarily by their Fe/Ca ratio, which tends to be lower in the latter. In Maronia, rare crystals have been identified with a general formula of (REE, Ca, Th)$_{3.97}$ Fe (Fe, Al, Ti)$_{1.8}$Ti$_2$(Si$_{2.1}$O$_7$)$_2$O$_8$ (Table 2). However, due to the absence of EPMA (Electron Probe Microanalysis) data, the lack of a systematic nomenclature scheme, the uncertainty of the Fe$^{+2}$/Fe$^{+3}$ ratio, and

the metamict nature of these minerals [65], it is currently not possible to assign a specific mineral name accurately. Therefore, based on the obtained Fe/Ca ratio and the observed dominance of Ce among the other REE, these minerals will be referred to as chevkinite-(Ce)-like phases. The chevkinite-(Ce)-like phases primarily occur as subhedral grains with diameters ranging from 3 to 5 μm. They are found either in contact with magmatic ilmenite or as inclusions in magmatic magnetite (Figure 5a,b). Based on this mode of occurrence, the chevkinite-(Ce)-like phase is considered to be of primary magmatic origin.

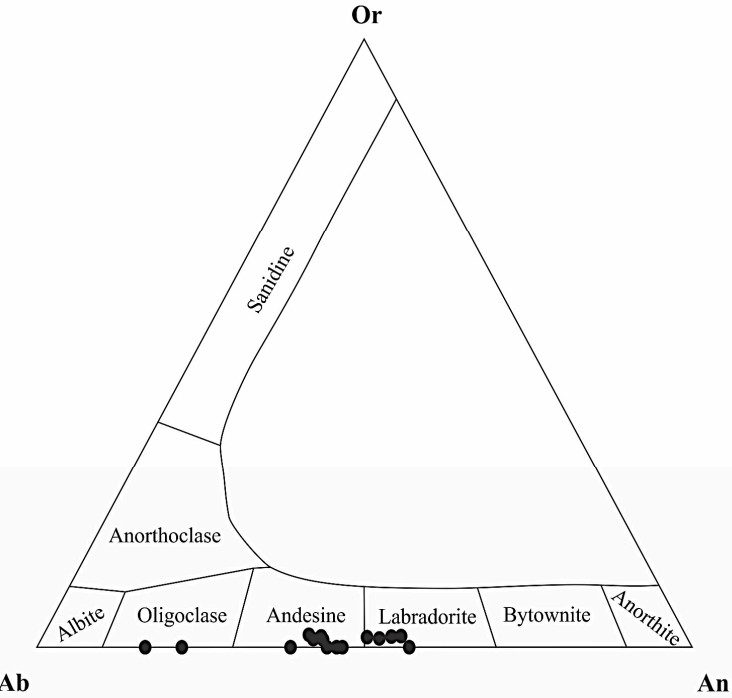

**Figure 4.** Ternary feldspar classification diagram showing the compositional range of analyzed plagioclase crystals in the Maronia monzodiorite (black circles).

**Table 2.** Representative mineral analyses of the accessory minerals and Fe-Ti oxides in Maronia monzodiorite.

| Sample | | | | | | | MAR-1 | | | | | | |
|---|---|---|---|---|---|---|---|---|---|---|---|---|---|
| Mineral | REE-ilm | Chv. phase | Aln | Mon | Hut.-Mon | Tht | Tht-cof | Bad | Zrl | Y-Zrl | Urn-Thn | Mn-Ilm | V-Mag |
| $SiO_2$ | n.d. | 22.5 | 34.6 | 1.60 | 14.1 | 18.2 | 17.9 | n.d. | n.d. | n.d. | n.d. | n.d. | n.d. |
| $TiO_2$ | 51.6 | 19.3 | n.d. | n.d. | n.d. | n.d. | n.d. | 1.28 | 34.3 | 34.9 | n.d. | 49.1 | n.d. |
| $Al_2O_3$ | n.d. | 3.17 | 19.2 | n.d. | n.d. | n.d. | n.d. | n.d. | n.d. | n.d. | n.d. | n.d. | n.d. |
| MnO | n.d. | n.d. | n.d. | n.d. | n.d. | n.d. | n.d. | n.d. | n.d. | n.d. | n.d. | n.d. | n.d. |
| MgO | n.d. | n.d. | 0.74 | n.d. | n.d. | n.d. | n.d. | n.d. | n.d. | n.d. | n.d. | n.d. | n.d. |
| FeO | 27.7 | 8.82 | 14.7 | n.d. | n.d. | n.d. | n.d. | 1.74 | 12.5 | 14.3 | 1.49 | 47.9 | 98.0 |
| $V_2O_3$ | n.d. | n.d. | n.d. | n.d. | n.d. | n.d. | n.d. | n.d. | n.d. | n.d. | n.d. | 3.05 | 2.00 |
| CaO | 5.92 | 6.04 | 10.9 | n.d. | 1.13 | n.d. | n.d. | 0.52 | 7.41 | 5.99 | n.d. | n.d. | n.d. |
| $Na_2O$ | n.d. | n.d. | n.d. | n.d. | n.d. | n.d. | n.d. | n.d. | n.d. | n.d. | n.d. | n.d. | n.d. |
| $K_2O$ | n.d. | n.d. | n.d. | n.d. | n.d. | n.d. | n.d. | n.d. | n.d. | n.d. | n.d. | n.d. | n.d. |
| $P_2O_5$ | n.d. | n.d. | n.d. | 30.8 | 8.57 | n.d. | n.d. | n.d. | n.d. | n.d. | n.d. | n.d. | n.d. |
| $La_2O_3$ | 4.20 | 11.3 | 6.11 | 25.0 | 6.30 | n.d. | n.d. | n.d. | n.d. | n.d. | n.d. | n.d. | n.d. |
| $Ce_2O_3$ | 3.35 | 21.4 | 11.1 | 34.0 | 9.44 | n.d. | n.d. | n.d. | 4.70 | n.d. | n.d. | n.d. | n.d. |
| $Nd_2O_3$ | n.d. | 7.46 | 2.61 | 6.10 | 2.88 | n.d. | n.d. | n.d. | 2.90 | n.d. | n.d. | n.d. | n.d. |
| $Y_2O_3$ | n.d. | n.d. | n.d. | n.d. | n.d. | n.d. | n.d. | n.d. | n.d. | 12.3 | n.d. | n.d. | n.d. |
| $ZrO_2$ | n.d. | 0.00 | n.d. | n.d. | n.d. | n.d. | n.d. | 96.5 | 23.4 | 32.4 | n.d. | n.d. | n.d. |
| $ThO_2$ | 1.25 | 0.00 | n.d. | 2.56 | 57.1 | 81.8 | 61.2 | n.d. | 2.38 | n.d. | 20.6 | n.d. | n.d. |
| $UO_2$ | 5.98 | n.d. | n.d. | n.d. | n.d. | n.d. | 20.9 | n.d. | 13.8 | n.d. | 77.9 | n.d. | n.d. |
| Total | 100 | 100 | 100 | 100 | 100 | 100 | 100 | 100 | 100 | 100 | 100 | 100 | 100 |

Notes: n.d.: not detected, Chv. phase: chevkinite-like phase, Aln: allanite, Mon: monazite, Hut.-Mon: huttonitic monazite, Tht: thorite, Tht-cof: thorite–coffinite, Bad: baddeleyite, Zrl: zirconolite, Y-zrl: yttrian zirconolite, Urn-Thn: Uraninite–thorianite, Mn-Ilm: Mn-bearing ilmenite, V-mag: vanadiferous magnetite.

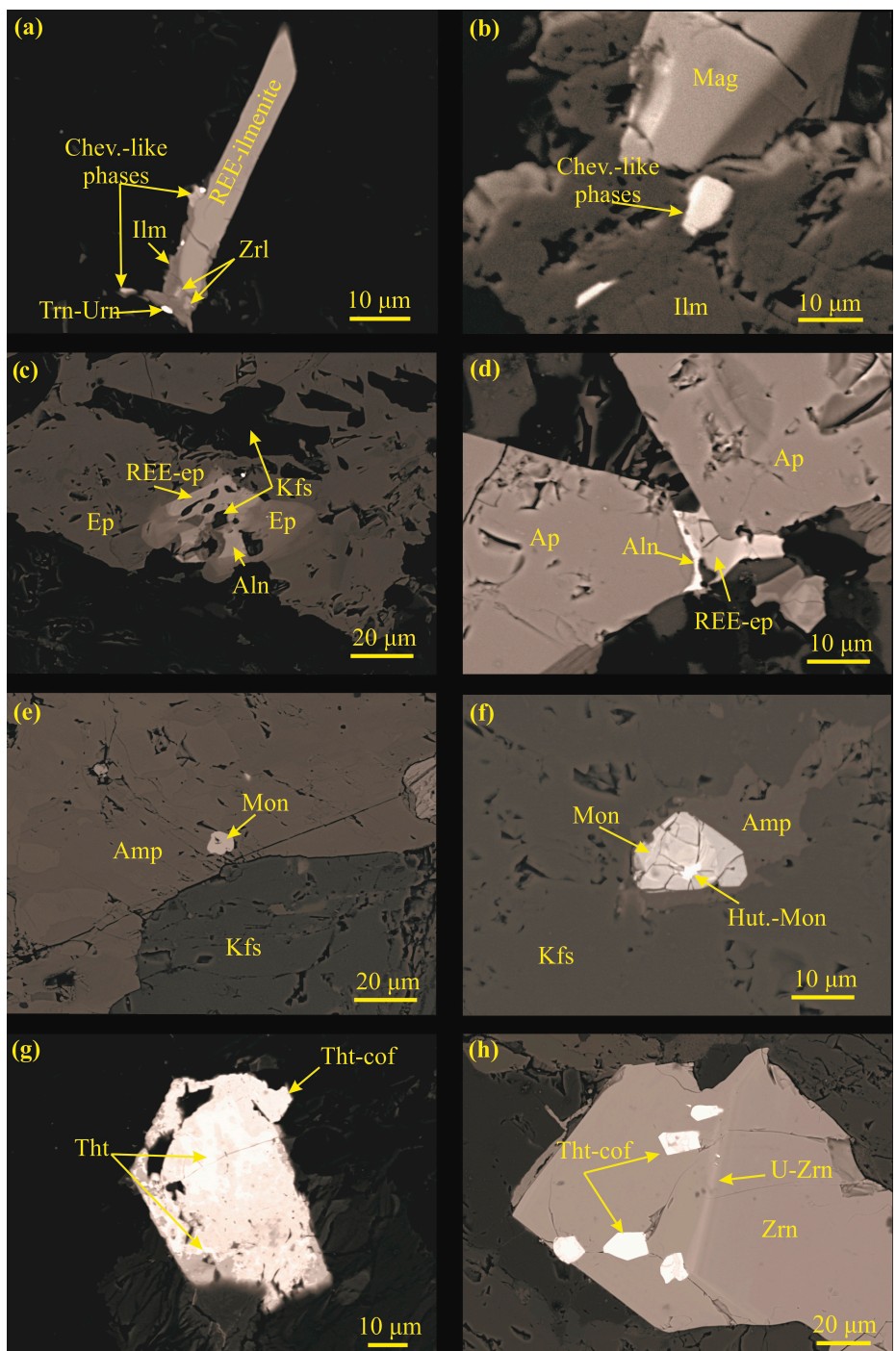

**Figure 5.** Back-scattered electron (BSE) images of chevkinite-like phases (**a**,**b**), early and late allanite-(Ce) (**c** and **d**, respectively), monazite-(Ce) and huttonitic monazite-(Ce) (**e** and **f**, respectively), and the associations of zircon–thorite–coffinite (**g**,**h**). Abbreviations: Chev.-like phases = chevkinite-like phases, Ilm = ilmenite, Zrl = zirconolite, Trn-Urn = uraninite–thorianite solid solution, Mag = magnetite, Aln = allanite, Ep = epidote, REE-ep = REE-bearing epidote, Kfs = K-feldspar, Ap = apatite, Amp = Amphibole, Mon = monazite, Hut.-Mon = huttonitic monazite, Tht = thorite, Tht-Cof = thorite–coffinite solid solutions, Zrn = zircon, U-Zrn = U-enriched zircon.

### 4.3.2. Allanite-(Ce)

Allanite, the REE-rich member of the epidote supergroup, is a commonly occurring accessory mineral in magmatic rocks. It is predominantly found in metaluminous and peralkaline granitic rocks, although its presence in gabbros and diorites is not uncommon ei-

ther [66,67]. In the Maronia samples, allanite has been observed in two modes of occurrence: (1) as independent euhedral-to-anhedral 20 μm diameter grains dispersed throughout the rock, and (2) as anhedral crystals bounded by fractures, or interstitially developed engulfing grain boundaries of major, rock-forming minerals (Figure 5c,d, respectively).

Early allanite may exhibit local dissolution features, and all allanite crystals may display chemical zoning, indicating a transition from allanite to REE-bearing epidotes. Based on these textural features, two generations of allanite can be identified in the Maronia monzodiorite: one formed as an early magmatic phase and the other formed as a late magmatic one. Unlike the chevkinite-(Ce)-like phases, allanite does not appear to be closely associated with any specific mineral. Chemically, both textural types of allanite show a dominant presence of Ce (Table 2) and are therefore identified as allanite-(Ce).

### 4.3.3. Monazite-(Ce) and Huttonitic Monazite

Monazite ((La, Ce, Nd) PO$_4$) is found as small subhedral crystals, primarily as inclusions within major rock-forming minerals, such as amphibole (Figure 5e). The chemical composition of the monazite crystals analyzed indicates a dominant presence of Ce, leading to their identification as monazite-(Ce). Additionally, these samples may contain elevated levels of Th, indicating the presence of a variable huttonite component. However, the analysis results reveal the existence of two distinct groups of REE phosphates: (1) monazite-(Ce) with a minor huttonite component, and (2) huttonite-dominated grains with a subordinate monazitic component (Figure 6, Table 2). Interlocking textures between huttonitic monazite and V-bearing magnetite are locally observed, suggesting a magmatic origin.

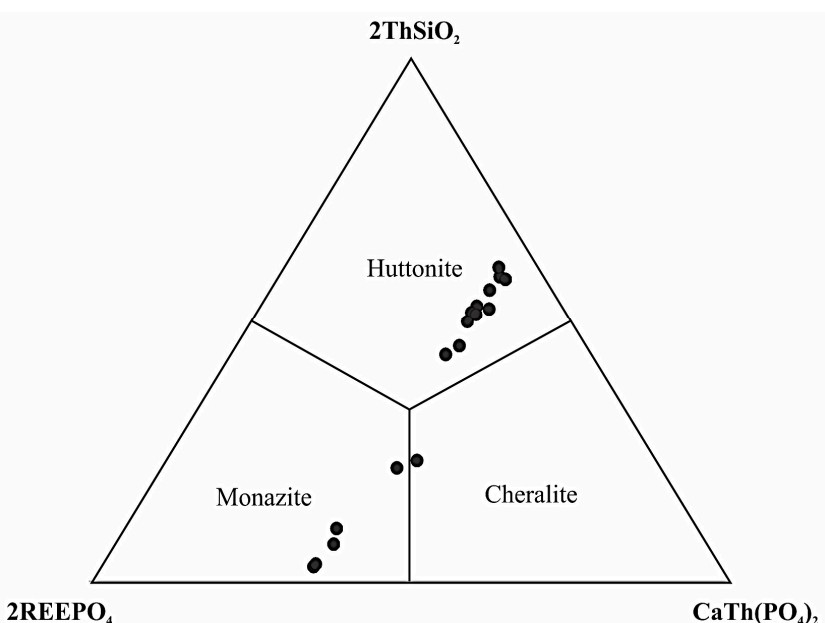

**Figure 6.** Ternary classification diagram of [68], showing the compositional variations in the analyzed REE-Th phosphate minerals (black circles) in Maronia monzodiorite.

In Maronia monzodiorite, huttonitic monazite occurs as a younger phase, filling fractures in pre-existing monazite-(Ce) (Figure 5f) or displaying textures that suggest replacement of the latter. Huttonitic monazite is also observed surrounding an apatite inclusion, within biotite. These heterogeneous huttonitic monazite-(Ce) grains are not in proximity to fractures or hydrothermal minerals; however, they appear to enclose magmatic V-bearing magnetite with apatite inclusions. Furthermore, they may cross grain boundaries, or traverse dissolution voids and cracks within the pre-existing rock-forming minerals. Therefore, it is evident from the above textural characteristics that the huttonitic monazite is younger than monazite-(Ce), as a late magmatic phase.

### 4.3.4. Thorite–Coffinite

In Maronia monzodiorite, thorite and coffinite (ThSiO$_4$-USiO$_4$) occur as solid solutions (Table 2) and appear, texturally, in two modes of occurrence: (1) inhomogeneous, independent, subhedral thorite crystals that are dispersed throughout the rock, and with dissolution features that exhibit replacement textures by coffinite, and (2) thorite inclusions in magmatic zircon (Figure 5g,h). In the latter case, thorite–coffinite appears subhedral and unfractured, whereas the host zircon is fractured. The thorite–coffinite crystals appear concentrated towards the outer zone of the host zircon, marked by a U-enriched zircon zone that runs parallel to crystal growth. These textural characteristics suggest that thorite–coffinite may represent a magmatic phase, formed concurrently with the ongoing growth of zircon following a magmatic event associated with U-saturation.

### 4.3.5. Baddeleyite

Baddeleyite is a mineral, the chemistry of which approaches almost stoichiometric ZrO$_2$. In Maronia monzodiorite, this mineral occurs as rare ~10 μm independent grains, dispersed throughout the sample, in proximity with either Mn-ilmenite or chloritized amphiboles (Figure 7a,b). Baddeleyite appears to be homogeneous and subhedral, lacking any discernible textural association with other Zr-bearing minerals such as zircon or zirconolite. From a chemical perspective, it deviates slightly from the ideal mineral formula due to the presence of minor amounts of Ti, Fe, Ca, and occasionally Th (Table 2). The ZrO$_2$ concentration varies within the range of 83.7 to 96.5 wt.%, resulting in an average mineral formula of $Zr_{0.91}(Ti, Fe, Ca, Th)_{0.09}O_2$.

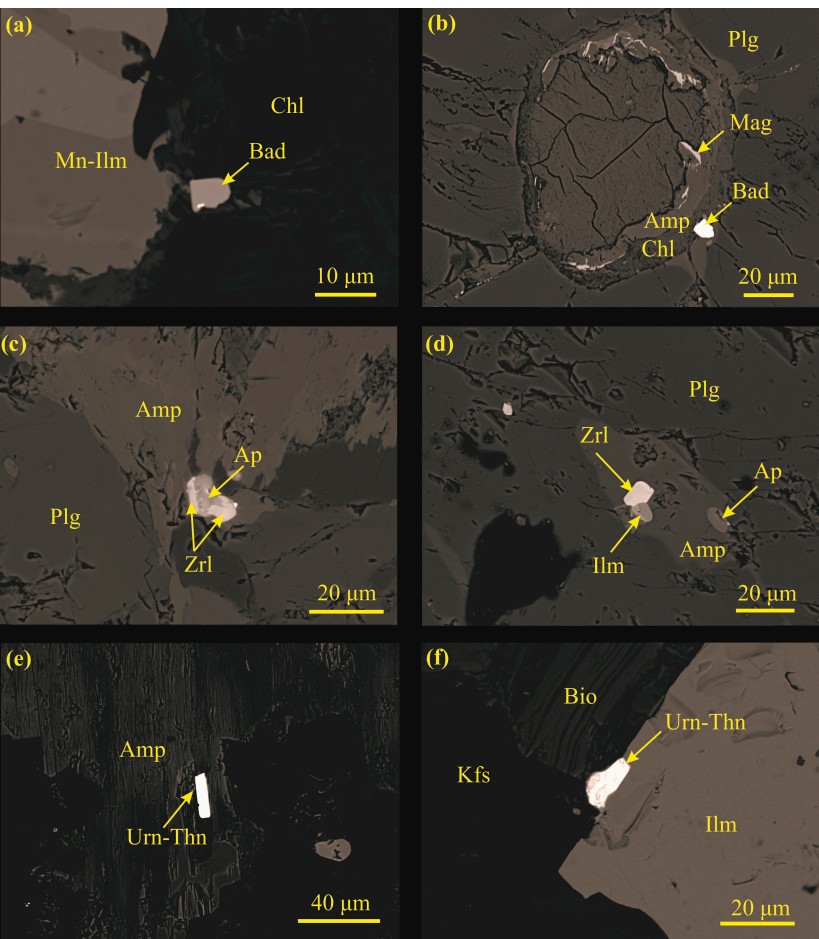

**Figure 7.** BSE images of Maronia monzodiorite showing the modes of occurrence for baddeleyite (**a**,**b**), zirconolite (**c**,**d**) and uraninite–thorianite solid solutions (**e**,**f**). Abbreviations: Mn-Ilm = Mn-bearing ilmenite, Chl = chlorite, Plg = plagioclase, Bio = biotite. All other abbreviations can be found in Figure 5.

### 4.3.6. Zirconolite

Zirconolite is a perovskite-type mineral with a general chemical formula of $CaZrTi_2O_7$, although its chemistry and structure have been reported to be quite variable as different polytypes of either monoclinic or trigonal symmetry have been discovered [69]. In Maronia monzodiorite, zirconolite has been observed in various textural modes, appearing as euhedral prismatic-to-anhedral crystals. It is found (a) interstitially between rock-forming minerals, (b) as overgrowths on apatite, or (c) mineral inclusions (Figure 7c,d). Zirconolite is closely associated with ilmenite, often encasing apatite, and found in proximity to chevkinite-like phases and Th-U oxides. Additionally, zirconolite has been identified as an inclusion within quartz, exhibiting dissolution features and being replaced by thorite.

As a metamict phase, the chemical composition of zirconolite can vary. However, two primary groups of zirconolite have been identified in the studied samples: (a) plain zirconolite and (b) yttrian zirconolite (containing up to 12.3 wt.% $Y_2O_3$, as shown in Table 2). Both types of zirconolite share similar textural characteristics, although yttrian zirconolite is relatively less common and is found as an interstitial phase between V-bearing magnetite and Mn-ilmenite with apatite inclusions. Yttrian zirconolite exhibits lower levels of Ca and Th compared to plain zirconolite. However, REE (up to 6 wt.% ΣREE) are predominantly incorporated into plain zirconolite.

### 4.3.7. Uraninite–Thorianite

In Maronia monzodiorite, several grains of U-Th oxides have been discovered and analyzed. These oxides are identified as solid solutions within the uraninite–thorianite system. They are observed as euhedral prismatic-to-anhedral inclusions within rock-forming phases such as quartz, biotite, and ilmenite (Figure 7e). Additionally, they are found as minute grains in contact with zirconolite and as interstitial phases between rock-forming minerals (Figure 7f).

The chemical analyses of these U-Th oxide minerals reveal a relatively consistent composition (Table 2), with an average content of 75 wt.% $UO_2$ and 25 wt.% $ThO_2$.

## 5. Discussion

### 5.1. Formation Conditions and Paragenetic Sequence of the REE-Zr-U-Th Minerals

Unlike major rock-forming phases, accessory minerals are highly sensitive to changes in physicochemical parameters. Consequently, associations of accessory minerals play a crucial role as indicators of magmatic and hydrothermal conditions. In recent years, the development of modern in situ analytical methods has allowed for better detection and more precise determination of these phases, significantly expanding our knowledge of their distribution and genetic interpretation [67].

The previously described REE-Zr-U-Th minerals from Maronia monzodiorite can be grouped into three main compositional categories: (a) the REE-Ti-Zr association, comprising REE-ilmenite, chevkinite-like phases, zirconolite, and baddeleyite; (b) the REE-Ca-P association, which includes early and late allanite-(Ce), monazite-(Ce), and huttonitic monazite-(Ce); and (c) the Th-U minerals, represented by thorite–coffinite and uraninite–thorianite.

The first assemblage is characterized by minerals low in Si, Ca, and Al, but enriched in Ti, Zr, and Fe when compared to the REE-Ca-P group. The concentrations of REE are generally lower in the REE-Ti-Zr group, with chevkinite-like phases being the principal hosts. Conversely, in the REE-Ca-P group, all minerals contain significant amounts of these elements. Interestingly, the actinides exhibit similar variations in both groups, suggesting that the incorporation of Th and U was likely influenced by their availability in the melt.

### 5.1.1. Formation of the REE-Ti-Zr Mineral Assemblage

In contrast to REE-bearing ilmenite (Table 2), which lacks available data in the literature, chevkinite-group minerals have been referred in various igneous compositions, ranging from alkaline and metaluminous granites and rhyolites [70–74], to intermediate and mafic compositions [74,75]. Chevkinite may occur as an accessory phase in metamor-

phic rocks and even constitute as an accessory mineral in metasomatic rocks [12,76]. While chevkinite is relatively rare in mafic compositions, its dimorph perrierite is more abundant [64]. Experimental data on the stability of chevkinite in igneous systems demonstrate that this mineral is an orthomagmatic phase (660–1050 °C), with elevated $Al_2O_3$ content possibly indicative of an increased crystallization pressure [77–79].

Compositionally, the Maronia chevkinite-like phases, most closely, resemble magmatic chevkinite, as shown in Figure 8 [80]. In the literature, the association of chevkinite and Ti-minerals has been generally described as a hydrothermal alteration, converting chevkinite to either ilmenite, rutile or titanite [64,72,81]. However, in Maronia monzodiorite, the textural evidence suggests otherwise. The REE-bearing ilmenite appears as a large prismatic, homogeneous and subhedral grain in contact with quartz and feldspar, indicating its primary magmatic origin. The crystallization of Fe-Ti oxides in Maronia pluton has been empirically estimated at 700 °C [54], confirming an orthomagmatic stage of formation. Chevkinite-like phases occur as patches within plain ilmenite that has itself developed replacing the REE-bearing ilmenite. Chevkinite-like phases occur as patches within plain ilmenite that replace the REE-bearing ilmenite. Considering the compositionally magmatic affinity of the chevkinite-like phase and the absence of hydrothermal alterations in these samples, we consider both REE-ilmenite and the chevkinite-like phase as orthomagmatic minerals. The latter formed at a later stage than the former, at a potential temperature range of 600–700 °C.

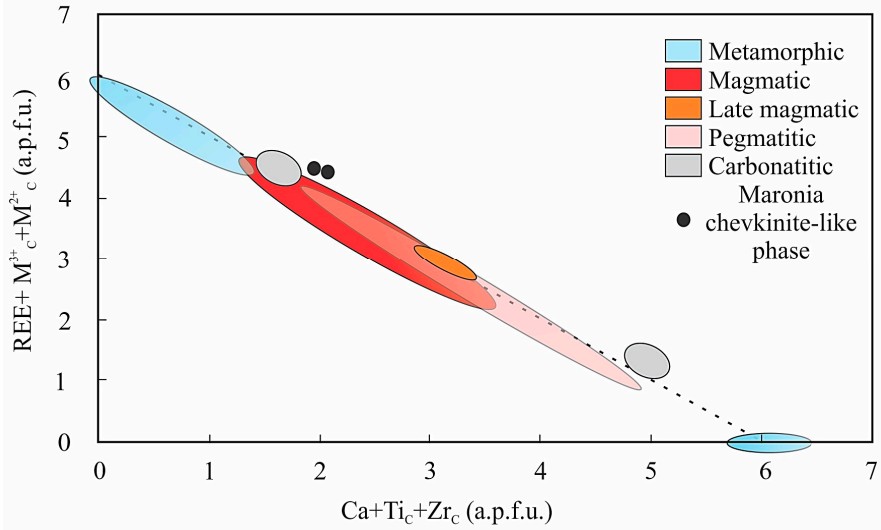

**Figure 8.** Chemical classification diagram showing the compositional ranges for the chevkinite–perrierite group of minerals, from different origins, modified from [80].

Chevkinite-like phases are often associated texturally with zirconolite, another Ti-bearing mineral. While zirconolite is more abundant than chevkinite, the presence of both minerals as inclusions in the same host grain (Figure 5a) suggests that the crystallization of zirconolite most likely occurred during the same magmatic stage.

The coupled behavior of Zr and Ti is indicated by traces of the former in some chevkinite-like phases and could be evidence of cationic substitutions between the two elements. Substitutions between Ti and Zr are also evident by the stoichiometry of zirconolite (Table S1). The M1-site shows only partial occupancy by Ca, REE, Th, and U; this suggests that Zr has been incorporated there as well, to account for the site deficit up to 1 a.p.f.u. This incorporation of Zr in the M1-site results in the partial filling of the M2-site (ideally fully occupied by Zr), where the remaining space is occupied by Ti. The Fe from the M4-site is incorporated in the M3-site, occupying the remaining space after the distribution of Ti cations. This stoichiometry suggests that during the formation of zirconolite, (a) Ca-deficiency led to its substitution by Zr, a characteristic that has been reported in natural zirconolites [82]; (b) Zr itself has been substituted by Ti in the M2-site;

and finally, (c) Ti has been substituted by Fe, which was highly abundant. The latter may imply the presence of both divalent and trivalent iron, the distribution of which among the M4 and M3 sites could be described by substitution # 4 of [82]: $2Fe^{3+} \leftrightarrow Ti^{4+}+Fe^{2+}$. Therefore, the excess of Fe in the M4 site presumably reflects the dominance of trivalent Fe, indicating possible crystallization under oxidizing conditions.

The retention of Zr in melts can be positively influenced by high magmatic temperatures and high alkalinity [83–85]. Additionally, experimental data have shown that the solubility of Zr in high temperatures is positively correlated with the presence of Ti in the melts [86]. Zirconium saturation temperatures in the Maronia monzodiorite were empirically determined using the Zr-saturation geothermometer of [86], calibrated for intermediate–mafic compositions. The temperature estimates fall within a range of 863 and 898 °C, higher than the crystallization temperatures of the Fe-Ti oxides in these rocks [54]. Thus, it is reasonable to assume that at least some of the observed Zr-bearing phases in Maronia monzodiorite crystallized at that temperature range. The coupled behavior of Ti-Zr in chevkinite-like phases and zirconolite, along with the occurrence of the latter as an inclusion in magmatic biotite, indicates that Zr was still available from the melt, at temperatures of 700 °C, close to the crystallization of Fe-Ti oxides, and 650 °C, prior to the crystallization of biotite [59].

Baddeleyite is the least abundant Zr-bearing phase in the Maronia monzodiorite. Compared to zircon, baddeleyite is Zr-dominated with traces of Ti and Fe in variable concentrations (Table 2). Zircon is more abundant and appears as large subhedral crystals in contact with major rock-forming minerals and is therefore a relatively earlier phase. Thus, it is inferred that zircon was the earliest phase of the Zr-saturation event (863–898 °C) prior to the Ti-Fe saturation (700 °C), during which both Ti and Fe were incorporated in zirconolite and baddeleyite. The strong association of zirconolite with Fe-Ti oxides implies the effect of localized Ti-Fe saturation on the crystallization of zirconolite. It is also noteworthy that from the inferred paragenetic sequence of the Zr-minerals, it appears that the earliest phase is a silicate mineral, whereas the subsequent phases are Si-deficit phases. This implies a limited Si availability in later magmatic stages during the formation of zirconolite and baddeleyite.

In summary, zircon is considered the earliest phase in the REE-Ti-Zr assemblage, with an estimated initiation of crystallization at temperatures between 898 and 863 °C. The crystallization of REE-ilmenite followed, along with Fe-Ti oxides at ~700 °C, which subsequently produced chevkinite-like phases and zirconolite, at a temperature range of 700–660 °C, under oxidizing conditions. Baddeleyite was the final mineral to be formed in the paragenetic sequence of the REE-Ti-Zr assemblage, at a later magmatic stage.

### 5.1.2. Formation of the REE-Ca-P Mineral Assemblage

Allanite-(Ce) can occur in various geological environments, including magmatic, hydrothermal, metamorphic, and metasomatic settings [15,86–88]. However, it is more frequently reported in intermediate-to-felsic, alkaline and metaluminous igneous rocks [69,70,89,90]. In mafic–intermediate rocks, allanite-(Ce) is relatively rare, although it has been identified in alkaline gabbros [91]. In the Maronia monzodiorite, allanite-(Ce) appears with distinct textural features, indicating two groups of this mineral with different relative ages: early and late magmatic (Figure 5c,d). All grains of early allanite-(Ce) exhibit zoning and dissolution features, suggesting that this early phase was not in equilibrium with the parent melt of the monzodiorite.

Allanite and monazite have similar crystallization temperatures in partial melts, yet monazite-(Ce) in Maronia monzodiorite does not appear as dissolved as the early allanite-(Ce) does (Figure 5e). A factor that may affect the relative stability field of these two minerals is the magmatic Si and Ca-activity [92]. However, the inclusions of dissolving allanite-(Ce) in major rock-forming minerals such as amphibole and plagioclase indicate an early origin in the monzodioritc melt. It is, therefore, possible that either (a) this early allanite-(Ce) constitutes a xenocrystic phase, since this mineral can be stable as a residual

phase, under at least 900 °C and 20 kbar, in crustal metamorphic rocks [87], or (b) magmatic conditions may have change rapidly after the crystallization of early allanite-(Ce).

The zoning of the early allanite-(Ce) in the Maronia monzodiorite suggests a progressive loss of REE, marked by the transition from a dissolved allanite core to epidote rims. Similar zoning has been reported in the allanite-(Ce) of the coastal black sands along the Kavala granodiorite that occurs at the same geological unit with Maronia in the Rhodope Massif [93]. The reverse is observed in the late allanite-(Ce), with a gradual increase in REE exhibited rimwards from an epidote core (Figure 5c,d). These textural features suggest that in early magmatic stages, a reduction in Ca-activity leads to the destabilization of early allanite-(Ce) and the resorption and removal of REE from that mineral. The incorporation of REE, at that stage, could be possible in the crystallizing and more-stable monazite-(Ce). Apatite in the monzodiorite does not contain REE. Apatite was most likely an orthomagmatic mineral, formed during or shortly after the crystallization of monazite-(Ce); thus, the available REE were incorporated in the latter rather than former mineral. Late allanite-(Ce), on the other hand, is found engulfing apatite grains (Figure 5d), and is, therefore, younger. This may indicate a subsequent increase of Ca-activity after the formation of monazite.

Potential ongoing changes in the magmatic Ca-activity are further indicated by the formation of huttonitic monazite-(Ce) in the Maronia monzodiorite. Huttonitic monazite-(Ce) has been found to replace a pre-existing apatite (Figure 7c) within a magmatic biotite grain. Huttonite is stable under mid-to-lower crustal pressures (12–30 kbar), and over a temperature range from 500 to 900 °C, with a stability field extending to even higher temperatures at lower crystallization pressures [94]. Maronia pluton is interpreted to have crystallized under upper crustal pressures of ~3–4.5 kbar [54] or ~1.1–1.5 kbar [58], suggesting that huttonitic monazite is a stable, high-temperature, magmatic phase, prior to the crystallization of the host biotite at ~650 °C [59] and after the crystallization of apatite. Thus, the formation of apatite and, subsequently, of late allanite-(Ce), may have removed Ca and REE, respectively, from the evolving melt, thus reducing the Ca-activity as to stabilize the huttonitic monazite. Additionally, it is also possible that the crystallization of zirconolite was synchronous with that of huttonitic monazite (i.e., under low Ca-activity), as evidenced by (a) its Ca-deficiency, (b) its variable concentrations of U and Th, and (c) its low REE concentrations (Table 2).

In conclusion, in the paragenetic sequence of the REE-Ca-P assemblage, early allanite-(Ce) is the oldest phase, the disequilibrium of which leads to the resorption and release of REE, followed by the formation of stable monazite-(Ce). A subsequent increase in the magmatic Ca-activity subsequently led to the formation of REE-free apatite followed by late allanite-(Ce). The crystallization of apatite and late allanite-(Ce) may have reduced the Ca-activity in the residual melt, stabilizing huttonitic monazite-(Ce) prior to the formation of magmatic biotite.

### 5.1.3. Formation of the Th-U Mineral Assemblage

The crystallization of magmatic zircon in Maronia monzodiorite has recorded a change in the solubility of U and Th, as demonstrated by a U-enriched zircon band, marking an outer zircon portion containing thorite–coffinite inclusions (Figure 5g). The silicate nature of thorite–coffinite suggests that the crystallization of these minerals occurred while and where Si was still available in the melt. On the other hand, uraninite–thorianite is often found in contact with Fe-Ti oxides and zirconolite and appears texturally the youngest (Figure 5a).

Based on the above textural evidence, it is inferred that Th-U saturation initiated soon after the onset of Zr-saturation formed the zircon cores. The anomalously high incorporation of U in zircon bands suggests that there are disequilibrium conditions between the growing zircon and the surrounding melt, leading to a sudden change in the distribution coefficient of U [93–96]. Based on their mode of occurrence, it appears that the U-Th silicates crystallized at earlier stages compared to the U-Th oxides. This is consistent with the

observed trend of Zr-minerals, which produced a silicate phase (i.e., zircon) at the early stages of Zr-saturation, followed by Si-deficit minerals (i.e., baddeleyite).

Thorianite and uraninite have mostly been reported from alkaline felsic rocks and interpreted as products of either high-temperature magmatic crystallization, low-temperature metasomatism, or hydrothermal alteration [97–99]. In mafic compositions, reports on thorianite and uraninite are sparce, although they have been recorded as metasomatic phases in ultramafic rocks [100]. The studied monzodiorite, however, is neither fractured, nor hydrothermally altered, suggesting that the observed Th-U minerals are of late magmatic origin, rather than hydrothermal. During the formation of zirconolite, oxidizing conditions could have prevailed, as suggested by its Fe-rich composition (Table 2) and its association with magmatic Fe-Ti oxides. Under oxidizing conditions, U could be in hexavalent form, as a uranyl ion; thus, U could be highly soluble and concentrated in the residual melt. A possible reduction mechanism of the uranyl ion has been proposed to be the presence of $S^{2-}$ from the oxidation of sulfide minerals, leading to its precipitation [97]. Both the chalcopyrite and pyrite of hydrothermal origin occur in the vicinity of the mineralized microgranite and are associated with the porphyry Cu-Mo-Au system [55,57]. The observed chalcopyrite and pyrite in the studied Maronia samples, however, are not fracture-bounded, nor in proximity with fractures or hydrothermal minerals, in order for an origin to be suggested. Furthermore, the location of the studied samples is more than one kilometer away from the outermost zone of the mineralized microgranite (Figure 1). The samples appear fresh with minimal indications of hydrothermal alteration (Figures S1 and S2); therefore, with the available textural data, we conclude that they are of late magmatic origin. In those samples, late magmatic pyrite and chalcopyrite have been replaced by Fe-oxides (Figure 9). Tetravalent, insoluble Th and U could have co-precipitated at that stage, and combined with a low Si-availability in the residual melt, produced solid solutions of uraninite–thorianite, as have been reported for magmatic Th-bearing uraninites [101].

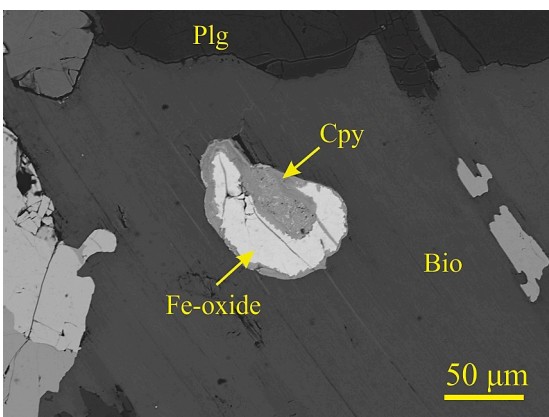

**Figure 9.** BSE image of a chalcopyrite (Cpy) inclusion in biotite (Bio) that has been replaced by Fe-oxides. Plg: plagioclase.

It is, therefore, possible that the U-Th saturation and precipitation in the monzodioritic melt initiated during the late stages of zircon crystallization while Si was still available, forming thorite–coffinite solid solutions. The formation of the U-Th oxides, however, could be the result of late crystallization under oxidizing conditions in a residual, Si-undersaturated melt. During this stage, the oxidation of late magmatic sulfides could lead to the reduction of the uranyl ion in the residual monzodioritic melt and the precipitation of tetravalent Th and U as uraninite–thorianite.

## 5.2. The Relationship between the REE-Ti-Zr-U-Th Minerals and the Evolution of Maronia Pluton

Geochemical and isotopic data from the Maronia mafic and intermediate rocks indicate that their parent melts originated from the low-degree partial melting of hydrous mantle sources, in the stability field of phlogopite, and were presumably emplaced at a depth of

~4–6 km [53,100]. The relative enrichment in critical metals of the Maronia shoshonitic magmas is considered to be the result of either (a) a metasomatically enriched mantle source, or (b) the assimilation of crustal material enriched in these elements [51,53]. Evidence of isotopic Sr-Nd, however, is interpreted as favoring an origin from a mantle source, enriched by subducted sediments and sediment-derived carbonatitic melts, rather than crustal assimilation [102,103]. A primary enrichment of the mantle source by subducted sediments is further supported by the high whole-rock $P_2O_5$ contents in Maronia monzodiorite [58], a geochemical feature associated with the melting of phosphate sediments in the source, and reflected by the presence of magmatic monazite-(Ce), apatite, and huttonitic monazite-(Ce), as described in this work.

The presence of zirconolite, baddeleyite, uraninite–thorianite and huttonitic monazite as accessory minerals in phlogopite-bearing peridotites has been considered to be the result of an anomalous magmatic enrichment in LREE and Zr by slab-derived, mantle metasomatism [98,103]. In those cases, the dominance of Th and U over REE, Nb and Ta, and of Fe over Mg, in the composition of zirconolite, is taken to reflect the presence of a Si-undersaturated, carbonatitic or meta-carbonate-derived melt in the source, presumably derived by slab melting [81,98,103]. Similar chemical affinities have been observed in the Maronia zirconolites (Figure 10), and together with the whole-rock shoshonitic affinity and isotopic characteristics of the parent melts, indicate that the enrichment in critical metals in Maronia monzodiorite could be the result of the partial melting of a hydrous mantle source, which was enriched with incompatible elements from phosphate and carbonate material by a subducting slab. The Ca-deficiency in zirconolite, however, does not indicate a Ca-poor metasomatic melt in the source, but is affected by the availability of Ca at the time of mineral formation.

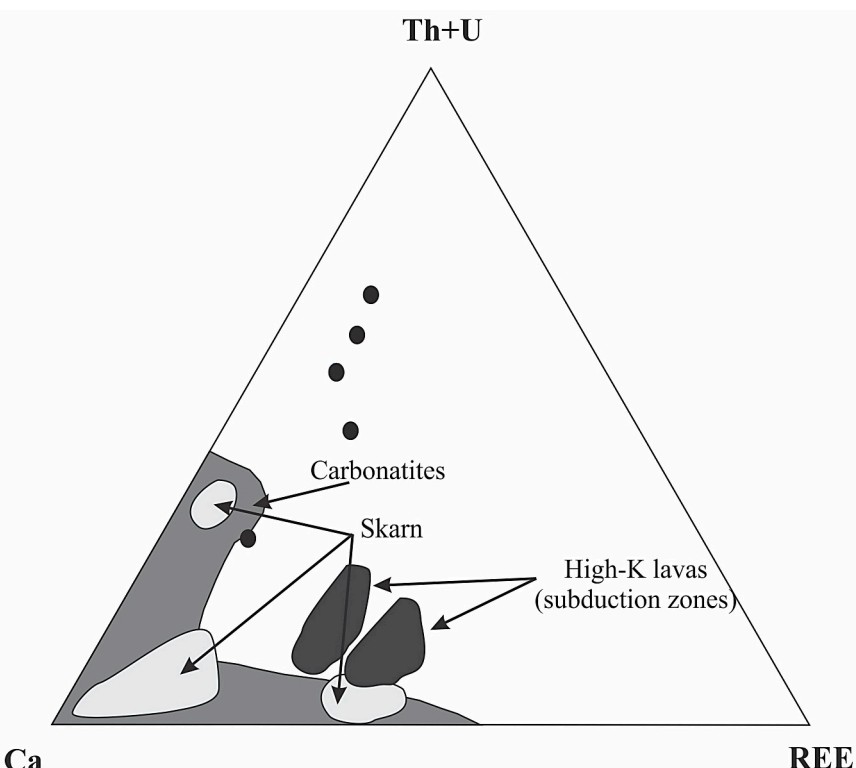

**Figure 10.** Ternary discrimination diagram showing the compositional ranges of natural zirconolites from different sources in ultramafic rocks [82,100]. The Maronia zirconolites are shown as black circles; elements are plotted as % atomic portions.

It is noteworthy that such a variety of REE-Zr-Ti-U-Th minerals occur in the monzodiorite Maronia Pluton, while their total whole-rock REE contents are no higher than 194 ppm (Table 1). The presence of several different Zr-minerals (i.e., zircon, zirconolite, baddeleyite)

and high Zr-saturation temperatures (>860 °C), while whole-rock concentrations of this element do not exceed 213 ppm (Table 1), may indicate that parameters other than melt composition lead to an oversaturation in critical metals in order to form their own distinct minerals. The alternating formation of allanite-(Ce) and monazite-(Ce) in the same location, as well as the change from silicate Zr-U-Th phases to oxides, demonstrates the presence of an open system, where Si and Ca activities probably change during the evolution of the melt. The Sr isotopic signature of the basic and intermediate groups in Maronia pluton points towards an open-system evolution [51], while the composition of plagioclase, which fluctuates from andesine to labradorite, with only limited normal zoning [51,55,57,58], suggests crystallization under the Ca variable's availability in the melt. Such changes have been associated with magma mixing and result in changes in the relative stability between monazite and allanite, while the Th increase in monazite has been associated with an increase in magmatic temperatures [104,105]. In potassic rocks, the presence of REE-Zr-Ti-Th minerals such as zirconolite, baddeleyite, thorite, thorianite and perrierite has been associated with the late magmatic oxidation of the parent mafic melts through mixing with fluids derived from felsic magmas [95,106,107].

Although Maronia Pluton is an inhomogeneous, composite intrusion that contains a microgranitic part, which itself hosts a Cu-Mo porphyry deposit [51,55,57], this felsic part appears younger than the main intrusion and is thus unclear whether it could be the source of the oxidizing fluids that affected the monzodioritic melt. In addition, the narrow range of low $SiO_2$ contents in the studied monzodiorite (52.6–53.2 wt%, Table 1) indicates a rather limited mixing with a high-Si component. It is more likely that fluids derived from the Maronia microgranite had a limited interaction with the monzodiorite and in subsolidus stages, as recorded by the presence of ore metals such as Pb, Cu and V filling cleavage planes and cracks in unaltered biotite (Figure 11a). On the other hand, an almost contemporaneous magmatism in the vicinity includes the 29.8 Ma, high Ti, P, Zr shoshonitic gabbro of Maronia, and the slightly older, 32–32.9 Ma, high-K, calc-alkaline melts in Kassiteres pluton, which presumably resulted in a different degree of partial melting of similar mantle sources [53,100]. A possible interaction of the Maronia monzodiorite with a less-evolved melt is indicated by the presence of late magmatic vanadiferous magnetite (0.7–2 wt.% $V_2O_3$, Table 2) that engulfs yttrian zirconolite and is in contact with Mn-ilmenite (Figure 11b). The formation of V-magnetite and Mn-ilmenite has been attributed to interactions with oxidized Fe-rich gabbroic magmas [108–111]. This demonstrates that the crystallization of zirconolite took place under the influence of an oxidized mafic melt. Based on the paragenetic sequence of the REE-Ti-Zr-U-Th minerals described above, it is evident that the magmatic conditions changed more than once during the evolution of the parent melt. Thus, it is probable that in this open magmatic system, an interaction with oxidizing mafic melts was ongoing, and re-equilibration continued until late magmatic stages.

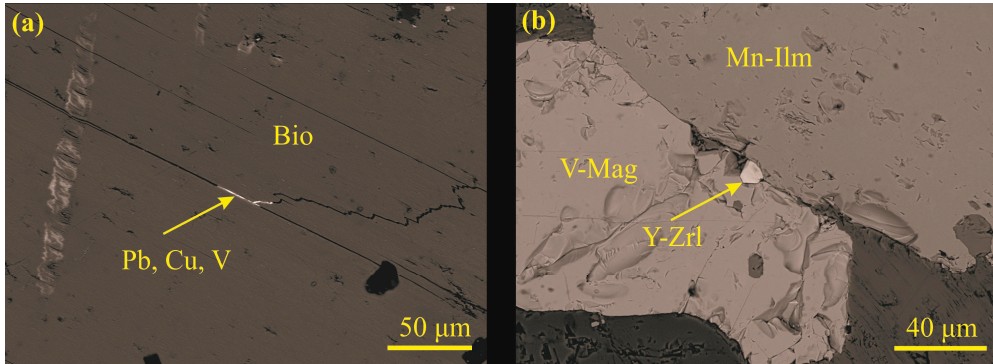

**Figure 11.** BSE images from Maronia monzodiorite showing (**a**) the subsolidus deposition of Pb, Cu and V along the cleavage planes and cracks in biotite (Bio) and (**b**) interstitial yttrian zirconolite (Y-Zrl) between vanadiferous magnetite (V-mag) and Mn-bearing ilmenite (Mn-Ilm).

It is therefore suggested, possible that the studied Maronia monzodiorite may have initiated as high-K, calc-alkaline partial melts, with similar affinities to those in the Kassiteres pluton. During early magmatic stages, a flux of younger, hotter, shoshonitic gabbroic melts (such as those of the Maronia basic group), derived from the slab-metasomatized mantle wedge, mixed with the evolving high-K, calc-alkaline magmas and re-equilibrated, as indicated by the variable plagioclase composition that mineralogically places these rocks at the boundary between monzogabbro and monzodiorite. In the initial high-K calc-alkaline melts, Ca-activity was presumably high enough to stabilize early allanite-(Ce). Mixing with the younger shoshonitic melts, however, reduced the Ca-activity and enriched the melt in P, Ti and Zr. This could have resulted in the resorption of early allanite-(Ce) and the subsequent formation of monazite-(Ce). This flux-derived Ti-Zr enrichment resulted also in the formation of the zircon, thorite–coffinite, chevkinite-like phases, and zirconolite in orthomagmatic stages under this low Ca-activity period. This assemblage formed under oxidizing conditions, as indicated by the presence of V-magnetite and Mn-ilmenite. Late magmatic allanite was formed subsequently, probably under increased Ca-activity; however a potential, subsequent, Si-Ca-undersaturation of the residual melt, may have resulted in the formation of huttonitic monazite instead, and oxidized late magmatic sulfides, reducing, thus, the solubility of U and Th. Therefore, at that late stage the forming minerals were, Si-deficit Zr-U-Th phases such as baddeleyite, thorianite and uraninite.

### 5.3. Implications on the Mineral Exploration of Critical Metals

Maronia pluton is an excellent case that demonstrates the essential role of geological processes in controlling the saturation and segregation of critical metals, even at bulk rock concentrations below those of economic interest. Ten different magmatic-to-late-magmatic minerals containing critical metals were identified in these mafic–intermediate , subduction-related rocks, with only limited primary enrichment. Thus, the insights gained from the detailed mineralogical and textural analyses of the REE-Ti-Zr-Th-U minerals in Maronia monzodiorite shed light on the control of specific subduction-related, geological processes for the mineral speciation of critical metals. These observations can have significant implications for mineral exploration strategies targeting critical metals, like REE, that could extend to mafic and intermediate igneous lithologies.

Understanding the complex formation processes and paragenetic sequences of minerals within specific geological settings is a fundamental guide for identifying the potential deposits of critical metals. The identification of distinct mineral assemblages associated with specific magmatic conditions in the Maronia monzodiorite demonstrates the importance of specific physicochemical parameters in enhancing REE to form their own distinct phases, rather than being incorporated as trace elements in other minerals. Such information is crucial in the design of exploration programs that may target specific geological environments more effectively. Geological processes in subduction settings, such as mantle metasomatism, are known to influence an enrichment in REE [86,112]. The geochemical change, however, from high-K, calk-alkaline to shoshonitic magmatism, may also be important for controlling the saturation of critical metals, leading to an enrichment of the produced igneous lithologies. Therefore, understanding the timing and conditions of critical metal mineralization can guide exploration efforts towards regions with analogous geological histories, improving the chances of discovering economically viable deposits in mafic–intermediate compositions.

The comprehensive study of mineral assemblages, paragenetic relationships, and textural features in complex geological settings, such as those in Maronia pluton, can provide insights into the processes governing critical metal mineralization and mineral speciation. This work demonstrates that in subduction-related systems, segregation of critical metals may be achieved. Applying these insights to exploration can lead to more targeted and efficient strategies, ultimately increasing the chances of discovering valuable deposits of critical metals in an extended compositional range.

## 6. Conclusions

Within the Maronia pluton, a comprehensive investigation into accessory phases within monzodiorite has unveiled an array of minerals significantly enriched in critical metals. These minerals fall into three primary groups: the REE-Ti-Zr, REE-Ca-P, and U-Th assemblages. The REE-Ti-Zr group includes REE-ilmenite, chevkinite-like phases, zirconolite, and baddeleyite. The second assemblage involves minerals like allanite-(Ce), monazite-(Ce), and huttonitic monazite-(Ce), while the third group comprises thorite–coffinite and uraninite–thorianite solid solutions. Through careful observations of textural features, the paragenetic sequence of these minerals was approached, enabling their correlation with the magmatic evolution of the host monzodiorite.

In the REE-Ti-Zr assemblage, the earliest mineral to crystallize is likely REE-ilmenite at temperatures around 700 °C. It is succeeded by chevkinite-like phases and zirconolite under oxidizing conditions, with baddeleyite emerging as a younger phase. In the REE-Ca-P assemblage, allanite-(Ce) forms as the initial mineral, appearing as inclusions in major minerals, accompanied by resorption features and a gradual loss in REE content. The subsequent progression involves the formation of monazite-(Ce), followed by late allanite-(Ce) in subsequent stages. Huttonitic monazite-(Ce) postdates monazite-(Ce) and is identified as the most recent phase in this group. In the third mineral group, textural evidence indicates that U-Th silicate phases, like thorite–coffinite, emerged during earlier magmatic stages shortly after the crystallization of magmatic zircon, at temperatures exceeding 800 °C, whereas uraninite–thorianite solid solutions are late magmatic.

The fluctuation in the formation of allanite-(Ce) and monazite-(Ce), even in the same sample, highlights variations in magmatic Ca-activity. During phases of elevated Ca-activity, allanite-(Ce) emerges as the predominant phase. Conversely, under conditions of reduced Ca-activity, phases such as monazite-(Ce), huttonitic monazite-(Ce), and zirconolite, distinguished by its Ca-deficiency, become stable. This latter stage, characterized by oxidizing conditions, is believed to be influenced by an Fe-rich, oxidized gabbroic melt, evident through Fe-enrichment in zirconolite and coexistent V-magnetite and Mn-ilmenite. The shift from Zr-U-Th silicate phases during early magmatic stages (zircon, thorite–coffinite) to oxide counterparts in later stages (baddeleyite, uraninite–thorianite) signifies Si-undersaturation in the evolving melt.

These mineral variations likely stem from an initial magmatic interaction between a high-K, calc-alkaline melt, where Ca-activity was significant in stabilizing allanite-(Ce), and an influx of shoshonitic gabbroic melt enriched in P, Zr, Ti, and REE. The latter originated from a metasomatized, hydrous mantle source within the stability field of phlogopite, enriched in critical metals through subducted, slab-derived material. Continued interaction and equilibration between the melts, persisting until late magmatic stages, led to the oxidation of magmatic sulfides and probably to a Si-undersaturation in the crystallizing melt. This resulted in the conversion of chalcopyrite and pyrite into Fe-oxides and reduced U-Th solubility, generating Si-deficit phases like uraninite–thorianite.

Comprehending these geological processes and their influence on the concentration and mineral speciation of critical metals holds substantial implications for the mineral exploration of these elements within mafic–intermediate compositions and analogous geological contexts.

**Supplementary Materials:** The following supporting information can be downloaded at: https://www.mdpi.com/article/10.3390/min13101256/s1, Table S1: Chemical analyses and detailed stoichiometric formula of the Maronia zirconolites. Figure S1: Hand specimen photographs of Maronia monzodiorite samples. Figure S2: Representative microphotographs of the studied Maronia samples from a petrographic microscope under crossed polarizers.

**Author Contributions:** Conceptualization, C.V. and A.P.; methodology, C.V. and A.P.; formal analysis, C.V. and A.P.; investigation, C.V. and A.P.; resources, C.V. and A.P.; data curation, C.V. and A.P.; validation, C.V. and A.P.; writing—original draft preparation, A.P.; writing—review and editing, C.V.

and A.P.; visualization, C.V. and A.P. All authors have read and agreed to the published version of the manuscript.

**Funding:** This research received no external funding.

**Institutional Review Board Statement:** Not applicable.

**Data Availability Statement:** All data supporting reported results have been included in the manuscript and in Supplementary Materials' file.

**Acknowledgments:** This work is dedicated to the memory of Panagiotis Mitropoulos who led our way in research in Economic Geology and Geochemistry and inspired his students and colleagues. The authors are grateful to the Wiener Laboratory of the American School of Classical Studies of Athens for their assistance during SEM-EDS analyses and their invaluable support in this work. Evaggelos Michailidis, scanning electron microscopy technician, is kindly thanked for his assistance with the SEM-EDS analyses at the Laboratory of Economic Geology and Geochemistry, National and Kapodistrian University of Athens. Finally, the authors are grateful to the Academic Editors and the two reviewers, whose comments and suggestions have greatly improved the clarity of this work.

**Conflicts of Interest:** The authors declare no conflict of interest.

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
