# Peer review of "The REE-Zr-U-Th Minerals of the Maronia Monzodiorite, N. Greece: Implications on the Saturation and Segregation Mechanisms of Critical Metals in Intermediate–Mafic Compositions"

_minerals, doi:10.3390/min13101256_

Round 1
Reviewer 1 Report
This is an interesting and informative manuscript, which describes the distribution and the chemistry of REE-Ti-Zr-U-Th minerals in the monzodiorite of Maronia, Greece. The authors have a good background of this subject, and present a comprehensive model for the relationship between these minerals and the evolution of the Maronia pluton. This article is of interest to the readers of Minerals and should be published with minor revisions.
In general, the study is well presented and explained. The English is fine, and the text (data, results and discussion) is well organized. Also, the figures and the diagrams are well prepared and are necessary for supporting the data. There are only a few points that need to be addressed. However, these points can be corrected quite easily. I therefore recommend this manuscript to be published after minor changes. My comments on specific parts of the manuscript are reported below.
Specific comments
1. Title and text: There is only one monzodiorite occurrence in Maronia and I do not understand why the authors use the term monzodiorites in the title and within the text. I recommend them to use the word “monzodiorite” everywhere.
2. Page 2, Line 72: The uncommon REE enrichment of the monzonite in Vathi of Kilkis that occurs in the same geological unit with Maronia, at the Serbomacedonian massif, could be also mentioned here: Stergiou, C.L.; Melfos, V.; Voudouris, P.; Spry, P.G.; Papadopoulou, L.; Chatzipetros, A.; Giouri, K.; Mavrogonatos, C.; Filippidis, A. The geology, geochemistry, and origin of the porphyry Cu-Au-(Mo) system at Vathi, Serbo-Macedonian Massif, Greece. Appl. Sci. 2021, 11, 479.
3. Page 2, Line 76: The Maronia pluton is Oligocene in age.
4. Page 4, Line 77: There is something wrong with the cited CBGA.
5. Page 2, Line 91: … of metamorphic core complexes…
6. Page 2, Line 94: …that represent continental and oceanic crusts, and asthenospheric mantle.
7. Page 3, Line 101: The citations 37 and 38 are too old, and I would recommend to be replaced by some more recent papers: the references 39, 40, and 41, and Kydonakis, K., Brun, J.-P., Sokoutis, D., 2015a. North Aegean core complexes, the gravity spreading of a thrust wedge. J. Geophys. Res. Solid Earth 120, 595-616.
8. Page 3, Line 116: The citation 46 is rather old, and I would recommend to be replaced by some more recent references: 43, 44, 45.
9. Page 3, Line 120: … within the RCC and CRB, ….
10. Page 3, Line 132: … and c) an acid group of granite and aplitic dykes and microgranite….
11. Page 3, Line 141: A comment about the porphyry Cu-Mo-Re-Au mineralization related to the microgranite, and that the molybdenite contains extreme Re content, can be added here, with the following citations: 55, 57.
12. Page 4, Line 146: The reference 100 should be cited also here.
13. Page 4, Line 155: The sites of the studied samples must be added on the map of Figure 1.
14. Page 5, Line 173: The change of paragraph is not necessary.
15. Page 6, Table 1: It would be more suitable for the readers, the table to be presented in one page. The Total of the major oxides and the ΣREE should be added, and the REE could be distinguished from the other trace elements, in the end of the Table.
16. Page 7, Lines 213-218: Definitely the mineralogical composition presented here is incorrect. The authors must check again their samples and also advise the references 51, 54, 100. Chalcopyrite and pyrite are probably of hydrothermal origin and this must be considered in the Discussion (Page 16, Line 542).
17. Page 7, Lines 222-223: The assumption that the plagioclase compositions, ranging from An40 to approximately An60, suggest that these rocks lie at the boundary between monzodiorite and monzogabbro, is not correct and has to be revised.
18. Page 8, Line 233: … like Fe; C consists….
19. Page 15, Lines 477-479: Similar zoning has been reported in the allanite-(Ce) of the coastal black sands along the Kavala granodiorite that occurs at the same geological unit with Maronia, in the Rhodope massif (Peristeridou E., Melfos V., Papadopoulou L., Kantiranis N., Voudouris P. (2022). Mineralogy and mineral chemistry of the REE-rich black sands in beaches of the Kavala district, Northern Greece. Geosciences, 12, 277, 21 p.), and can be cited here.
20. Page 16, Lines 496-497: Maronia pluton is interpreted to have crystallized under upper crustal pressures of ~3-4.5 kbar [54], or ~1.1-1.5 kbar [100], ….
21. Page 16, Lines 540-542: Chalcopyrite and pyrite have possibly a hydrothermal origin and definitely their replacement is supergene not related to any late magmatic reaction. This must be revised.
22. Page 18, Line 607: … [51,55,57,100]…
Author Response
A point-by-point response to the reviewer’s comments may be found in the file attached.

Reviewer 2 Report
The manuscript reports an interesting mineralogical and geochemical investigation on Maronia monzodiorite of Greece, providing a genetic model for the precipitation of rare minerals containing REE, Th, U and Zr. Furthermore, the Authors also discuss the potential economic implication of this type of mineralization. The manuscript is well organized, the data are important and, although I am not a native English speaker, the language is good and well understandable. Therefore, I consider the manuscript suitable for publication in Minerals after minor corrections. Here are some suggestions that I hope will help the Authors to improve their manuscript.
1) the Authors use two different acronyms to define the rare earth elements. i.e. REE and REEs. Please select and use only one.
2) Line 52. Define the meaning of the acronym IOCG.
3) Line 77. Explain the meaning of the text written in Greek and CBGA.
4) Figure 1. Is not easy to recognize the difference of the grey colour.
5) Line 181. Remove rare earth elements, REE is enough.
6) Figure 3. No correlation are visible from these diagrams.
7) Table 1. The content of Cr2O3 is very low. Better to calculate it as Cr in ppm and insert it in the list of trace elements.
8) Line 208. I guess that not detected is not necessary.
9) Table 2. The totals are written in two rows and is not easy to read the abbreviation of the minerals. I suggest to plot the table in a landscape page or to reduce the characters.
10) Figure 8. The plot of only to analyses is, in my opinion, not enough to be sure of the meaning of the diagram.
11) Line 438. Baddeleyite.
12) Figures 5, 7, 9, 11. In the captions, please specify all the abbreviation of the minerals.
13) Use italic in all the title of subtitle of the chapters.
With my best regards.
Author Response

(The authors gave the same response as above.)
